# Associations between Daily Step Counts and Physical Fitness in Preschool Children

**DOI:** 10.3390/jcm9010163

**Published:** 2020-01-07

**Authors:** Chunyi Fang, Jinming Zhang, Tang Zhou, Longkai Li, Yaofei Lu, Zan Gao, Minghui Quan

**Affiliations:** 1School of Kinesiology, Shanghai University of Sport, Shanghai 200438, China; 2College of Sport Medicine and Rehabilitation, Shandong First Medical University & Shandong Academy of Medical Sciences, Taian 271016, China; 3School of Kinesiology, University of Minnesota-Twin Cities, Minneapolis, MN 55455, USA

**Keywords:** accelerometer, health outcome, physical activity, walking, young children

## Abstract

Purpose: To investigate the relationships between daily step counts and physical fitness in preschool children. Methods: Preschoolers’ step counts were assessed by ActiGraph accelerometers consecutively for seven days. Physical fitness was assessed by a 20 m shuttle run test (cardiorespiratory fitness), the handgrip and standing long jump tests (musculoskeletal fitness), and the 2 × 10 m shuttle run test (speed/agility). A composite score was created from the mean of the standardized values of all physical fitness tests. Results: A total of 301 preschoolers (134 girls, mean age 57.40 ± 5.47 months; 167 boys, mean age 58.10 ± 5.34 months) were included in the final analysis. Compared with the lowest tertile, boys and girls in the highest tertile of step counts achieved high physical fitness with odds ratio (OR) being 5.39 (95% CI = 1.65–17.59) and 4.42 (95% CI = 1.30–14.99), respectively, after adjusting for confounders. Meanwhile, a relationship was observed for each 1000 steps/day increment being associated with 43% (OR = 1.43, 95% CI = 1.10–1.85) and 62% (OR = 1.62, 95% CI = 1.20–2.19) increment for high physical fitness in boys and girls, respectively. In addition, significant non-linear relationship was observed between daily steps and physical fitness in boys, which indicated that accumulated 8000 steps/day was associated with the highest ratio to achieve high physical fitness. Conclusions: Positive relationships between step counts and physical fitness were observed in preschool children, and the relationships were strongest for those who accumulated 8000 steps/day in boys. To confirm the findings in this study, well-designed and large-scale longitudinal studies are needed in the future.

## 1. Introduction

Physical fitness refers to the ability to engage in daily physical activity (PA) without undue fatigue and the capability to cope with emergency situations and to improve health [1], which is considered as a combination of cardiorespiratory fitness (CRF), musculoskeletal fitness, and motor fitness in preschool children [2]. Previous studies have shown that high levels of physical fitness can help to maintain the optimal health status and lower a wide variety of disease morbidities and mortalities [3,4,5]. For example, cardiorespiratory fitness (CRF) [3] and musculoskeletal fitness [6] have been found to play positive roles in cardiovascular disease (CVD) prevention, and speed/agility is beneficial to skeletal health [7]. In addition, CRF is also positively associated with mental health and academic performance in children [7,8]. Moreover, children’s early years, of age three to six, is a vital window in time for physical fitness development [2], as high levels of physical fitness not only promote normal growth but also exert carryover effects that affect personal health and quality of life later in adulthood [2,9]. Hence, improving and maintaining physical fitness levels in preschool children becomes a priority issue around the world.

PA is regarded as a basic component of healthy lifestyles [10], which facilitates the preschooler’s CRF, musculoskeletal fitness, and motor fitness [11,12,13]. To achieve health benefits from regular PA participation, the U.S. [14], Canada [15], Australia [16], and the World Health Organization (WHO) [17] have issued specific PA guidelines for preschool children, which have stated that preschoolers should engage in ≥ 180 min per day in PA of all intensity levels, including ≥60 min of moderate-to-vigorous physical activity (MVPA). Although these guidelines offer a specific PA targets for preschool children, there is an obvious shortcoming that MVPA is difficult for the general population to comprehend and monitor. Therefore, the large-scale application of these guidelines is limited.

Walking is a predominant form of PA in many daily activities, and any assessment of PA should be sensitive for walking indicator—step count [18], an outcome highly correlated with MVPA [19] and easy to adopt and monitor. In this sense, in order to make it easier to facilitate health practitioners and researchers monitor PA in preschoolers, some studies aimed at transferring 60 min MVPA into the how many steps are needed per day [20,21,22]. In adult, previous studies indicated that daily step counts have been shown to significantly increase physical fitness levels [23] and reduce the risk of cardiovascular disease [24]. However, the associations between daily step counts and physical fitness in preschool children remain largely unexplored [14]. Therefore, the current study aimed to examine the relationships between daily step counts and physical fitness in preschoolers. Findings of this study may serve as an evidence-based foundation for use of daily step counts to promote physical fitness in preschool children. We hypothesized that preschoolers’ step counts would be positively correlated with physical fitness.

## 2. Methods

### 2.1. Participants and Research Setting

This cross-sectional study comprised of 401 preschoolers. For feasibility considerations, we consecutively recruited 7 preschools located in Baoshan and Yangpu Districts of Shanghai, China by using a convenient sample. The parents’ meeting was organized by the preschools to carry out project lectures and participants recruitment. The inclusion criteria of this study were: (1) children aged 3 to 6 years; (2) healthy, and without CVD, neurological or endocrine disease, which may hamper children to perform the fitness tests [25]; and (3) children’s parents or legal guardians provided informed consent. The study was approved by the Ethics Committee of Shanghai University of Sport (ethic committee code: 2015028).

### 2.2. Measurements

Considering the feasibility of measurement, data collection session was conducted only from 09:00 to 11:00 every weekday by four experienced researchers in kindergarten. Each group had eight children, and all the measurements were performed by groups. Those who waited for the measurement were in the classroom for the regular kindergarten curriculum.

Height in cm and weight in kg were measured according to the National Physical Fitness Measurement Standards Manual (preschool children version) [26]. Body mass index (BMI) was calculated as weight/height squared (kg/m^2^).

Daily step counts were assessed by using ActiGraph GT3X+ accelerometers (ActiGraph LLC, Pensacola, FL, USA) for 7 consecutive days with an adjustable elastic belt on the right hip while awake, except for bathing and swimming. The accelerometers have been recognized as technically reliable instruments to objectively capture daily step counts [27]. The sampling period was one second epoch [28]. Daily step counts lower than 1000 and above 30,000 per day were considered as invalid data [29]. Researchers suggested that 3 days is acceptable to monitor preschool children’s PA levels [30], and significant differences in PA amount between weekday and weekend day was found in previous studies [31,32,33]. Therefore, preschool children with valid days up to 2 weekdays and 1 weekend day were included in final analyses, which was also commonly used in previous studies [31,33,34].

Physical fitness assessments were performed according to the validated PREFIT battery [2], as evidenced by previous publications [13,35,36]. The details of each testing item were described as below:(1)CRF was assessed by the 20 m shuttle run test (20mSRT) [37]—a reliable assessment of CRF in preschool children [2]. During the test, children were asked to run in a straight line back and forth from the beginning point to other line (20 m apart) with the initial speed of 8.5 km/h and increased by 0.5 km/h/min. In order to maintain the speed, one researcher ran with one child per time and children were required to follow the researcher’s pace. Maximal performance was recorded when children failed to follow the pace for two consecutive occasions or stopped due to exhaustion [38]. The test was conducted once, and scores were expressed as stages (laps).(2)Musculoskeletal fitness was assessed by the handgrip test and standing long jump test (SLJ), which represented maximum handgrip strength and lower limb explosive strength, respectively [2]. The handgrip test was performed by using a T.K.K.5401 (Takei, Niigata, Japan, Tokyo), and the grip length was modified to the best position based on the length of the participant’s hand. As for the handgrip test, children were asked to stand on both feet and the tested arm straight down with the shoulder abducted about 10° without body touch. The elbow and wrist extended fully and forearm in neutral position [39]. With regard to SLJ, children were asked to stand behind a line, jump as far as they can with their feet together and keep their low limb upright when performing SLJ [40]. Researchers conducted an example before tests to make sure they are able to perform correctly. Each participant had two attempts and the better score was recorded in kilograms (precision of 0.1 kg) and centimeters (precision of 0.1 cm), respectively.(3)Motor fitness mainly consists of speed, agility and balance, which was considered as one of major component of physical fitness in early childhood [2]. It was assessed by the 2 × 10 m shuttle run test (10mSRT), which is also a very important testing item for Chinese National Preschool Children Physical Fitness Standard battery to assess the speed/agility since 2003 [26]. Participants were instructed to run as fast as possible from the beginning point to the other line (10 m apart) back and forth, which covered a distance of 20 m (2 × 10 m) in total. The test was performed twice, and the better score was used for data analysis (precision of 0.1 s).

### 2.3. Data Analysis

Participant characteristics are presented as mean ± SD for continuous variables and percentages for categorical variable across tertiles of daily step counts. Differences among daily step counts tertiles were analyzed by one-way ANOVA plus LSD post-hoc analysis, and Chi-square test for continuous and categorical variables in boys and girls, respectively. Statistically, daily step counts values being out of the range of standard deviations ±3.00 were considered as outliers and thus removed from the final analysis. Two sets of multivariate models were applied to investigate the relationships between preschoolers’ step counts and physical fitness. Model 1 adjusted for no variables and Model 2 adjusted for potential confounding variables like BMI, age, and the accelerometer wear time [41,42].

Due to sex-related differences in physical fitness [42], we stratified the data by sex. Physical fitness outcomes were transformed into standardized scores (i.e., Z-scores) due to the different arithmetic scales and units, standard value = (value-mean)/standard deviation. Associations between daily step counts and physical fitness were investigated by three steps. First, we multiplied the standardized value for 10mSRT score by –1, because it is inversely associated with speed /agility. The comprehensive Z-score of physical fitness was calculated as the mean of 4 standardized scores in different sex groups, respectively (Z_physical fitness_ = (Z_20mMSRT_ + Z_grip_ + Z_SLJ_ + Z_10mSRT_)/4). In addition, daily step counts were ranged from low to high and categorized into tertiles (T1–T3), and Z_physical fitness_ were categorized into quartiles (Q1–Q4). The Z_physical fitness_ was the dependent variable and expressed as a binary outcome with the highest quartile of Z_physical fitness_ (Q4) defined as high physical fitness and assigned a value of “1”, while other quartiles (Q1–Q3) were defined as non-high physical fitness and assigned a value of “0”. Second, we entered tertiles of step counts as the independent variable, comprehensive Z-score of physical fitness as a dependent variable, using logistic regression to examine the relationship between daily step counts and high physical fitness with the lowest tertile as reference. In addition, linear regression was applied to estimate the high physical fitness odds ratios (ORs) change followed by per 1000 steps per day increment. Third, we examined whether the associations between daily step counts and high physical fitness level vary with different daily step counts by using two-piecewise linear regression models.

The results from the logistic regression were presented as ORs and 95% confidence intervals (CIs). All data analyses were performed by using Empower (R) (www.empowerstats.com, X&Y solutions, Inc., Boston, MA, USA) and R (http://www.R-project.org). Statistical significance level was set at *p* < 0.05.

## 3. Results

A total of 301 participants from 401 preschoolers were included in our study. Of them, 100 participants were excluded in final data analysis due to the following reasons: (1) unable to participate in fitness test (*n* = 28); (2) valid accelerometer wear days did not yield at least 2 weekdays and 1 weekend day (*n* = 64); (3) failed to complete SLJ, grip strength, 20mSRT, or 10 mSRT test (*n* = 6); and (4) outliers (*n* = 2). Collectively, 301 children (167 boys and 134 girls, mean age 57.8 ± 5.4 months) were included in the statistical analysis (See Figure 1).

The characteristics of participants are expressed in mean ± SD according to step counts tertiles. Three characteristics of weight, BMI, and accelerometer wear time in boys showed significant differences among tertiles, while five characteristics of height, weight, grip, 20m SRT, 10mSRT, and accelerometer wear time in girls showed significant difference among tertiles. There were no significant differences in other characteristics among tertiles (See Table 1).

Children with higher daily step counts were more likely to achieve the high physical fitness level after adjusting for age, BMI, and accelerometer wear time. Compared with those of the lowest tertile of daily step counts, boys and girls in the highest tertile of daily step counts achieved high physical fitness with OR being 5.39 (95% CI = 1.65–17.59) (*p* for trend = 0.01) and 4.42 (95% CI = 1.30–14.99) (*p* for trend < 0.02), respectively, after adjusting for age, BMI, and accelerometer wear time. Meanwhile, a relationship was observed with an OR of achieving high physical fitness increased by 43% (OR = 1.43, 95% CI = 1.10–1.85) and 62% (OR = 1.62, 95% CI = 1.20–2.19) for every 1000 steps/day increment in boys and girls, respectively (See Table 2).

Further, a curvilinear relationship between step counts and high physical fitness level was observed in boys but not in girls after adjusting for age, BMI, and accelerometer wear time (See Figure 2). The high physical fitness OR in boys increased 1.85 times (OR = 2.85, 95% CI = 1.29–6.32) for every 1000 steps/day increment for preschoolers whose step counts below 8000 steps/day, after which the OR leveled (OR = 1.04, 95% CI = 0.70–1.55) (See Table 3).

## 4. Discussion

### 4.1. Main Findings in This Study

The aim of our study was to investigate the associations between preschoolers’ daily step counts and physical fitness. We found positive relationships between step counts and high physical fitness level. Moreover, the relationships were strongest for individual who accumulated 8000 steps/day in boys, but not in girls.

### 4.2. Comparison with the Findings from the Previous Publications

Our findings support the research hypothesis, and are with previous studies indicating that PA was significantly associated with physical fitness in preschool children. Data from cross-sectional studies suggested active children were more likely to have better physical fitness than less active [11] and sedentary behavior children [43]. Vigorous PA was also found beneficial to CRF, as children who accumulated 40 min vigorous PA was greater than those only had 18 min [44]. Further, longitudinal studies showed PA was positively correlated with CRF, musculoskeletal fitness, and motor fitness in preschool children [35,45]. To date, no empirical study has examined the associations between daily step counts and physical fitness in preschool children. Therefore, this study provides additional information and sheds a new light in the field of PA and health promotion.

Although step counts cannot reflect PA intensity well (current guidelines emphasize the importance of intensity), it is still an excellent indicator of health-related outcomes [46,47]. For example, longitudinal studies found that using daily step counts as a PA indicator can promote inactive individuals to accumulate more steps per day, which facilitate greater insulin sensitivity and lower adiposity [46], and stepping behavior was also found negatively correlated with CVD risk [48]. In addition, compared with MVPA targets in current PA guidelines, step counts can be more easily to be measured (i.e., pedometer and cell phone) and understood by general population, and hence it is more likely to be used with general population. Second, previous studies have shown step counts were positively correlated with MVPA [19], and our data also showed significantly correlations between them (*r* = 0.83, *p* < 0.001). Third, studies had found that even low intensity PA can also promote physical fitness [49]. Finally, the latest longitudinal study found that step counts were significantly associated with all-cause mortality rather than stepping intensity [50]. Hence, daily step counts could be regarded as an important indicator of physical fitness.

### 4.3. How Many Daily Steps are Enough for Preschool Children?

At present, it remains unknown how many steps per day are enough for preschool children to achieve optimal health status, although previous studies have primarily focused on step counts targets corresponding to PA guidelines. However, available literature was inconsistent and inconclusive. For example, cross-sectional findings from Gabel et al. [20] reported a step counts target of 6000 steps/day as equivalent to performing all intensities of activities at the recommended 180 min per day, of which included at least 60 min MVPA. Nevertheless, Vale et al. [21] recommended the step counts threshold of 9000 steps/day as equivalent to the 180 min PA recommendation performed at any PA intensity. One possible explanation for large gap from the available data may be due to the accelerometer cut-point non-equivalence used to assign MVPA (i.e., MVPA defined as ≥ 1680 versus ≥ 3200 counts/min) [51]. Because of the varied cut points setting, the mean measured time in MVPA ranged from 39.5 to 269 min per day [51], which may have led the inconsistent daily step counts recommendations. Obviously, findings from the aforementioned studies that based-on PA guidelines may not be comparable with our study, which was based-on physical fitness outcomes. To our knowledge, studies related to the associations between step counts and health-related outcomes in preschool children are scarce. However, the daily steps recommendation based on physical fitness outcomes should be more reasonable than the results based on PA guidelines. Therefore, further studies with well-designed and large-scale are needed to confirm the findings of this study.

Moreover, the association between daily step counts and high physical fitness was strongest for individual who accumulated 8000 steps/day in boys, but not in girls. Although the exact reason was not clear, a potential explanation regarding to this observed difference may be attributed to the relatively small sample size of girls, making it difficult to identify the true relationships between daily step counts and physical fitness. Therefore, future studies with more female participants are needed.

### 4.4. Potential Mechanisms

Several potential mechanisms could explain the associations between step counts and physical fitness. First, PA promotes mitochondrial biogenesis [52], facilitates mitochondria quantity and quality [53] and enhances aerobic capacity [53,54,55]. Second, PA increases the capillary-to-fiber ratio and capillary area as well as blood flow in target organs, which improves CRF and musculoskeletal fitness [56]. Third, PA regulates the skeletal muscle gene expression patterns [57], switches skeletal muscle fiber type, increases muscle mass, and regulates energy metabolism. Taken together, the findings of these studies provide evidence that PA could improve physical fitness via many pathways.

### 4.5. Strengths and Limitations

This study has several strengths. First, to the best of our knowledge, this is the first study to identify the associations between daily step counts and physical fitness in preschool children. Second, our findings were strengthened by both categorical and linear analysis. Some limitations, however, should be noted. First, this study used the cross-sectional study design, which has limited to draw a cause-effect relationship in findings. We cannot explain whether engaging in more daily step counts will improve physical fitness or if individuals with high level of physical fitness tend to accumulate more daily step counts. Second, a convenience sample was recruited from northeast area of Shanghai in the present study, which limited the generalizability of our findings among the populations. Moreover, findings of this study need to be interpreted with caution for the wide confidence intervals. Consequently, further investigations with representative and large sample size are warranted in future study.

## 5. Conclusions

In conclusion, we identified meaningful and positive relationships between daily step counts and physical fitness in preschool children. Furthermore, relationships were strongest for those who accumulated 8000 steps/day in boys. On the basis of this study, daily steps as an indicator for physical fitness should be considered in preschool children. To confirm the findings in this study, well-designed and large-scale longitudinal studies are needed in the future.

## Figures and Tables

**Figure 1 jcm-09-00163-f001:**
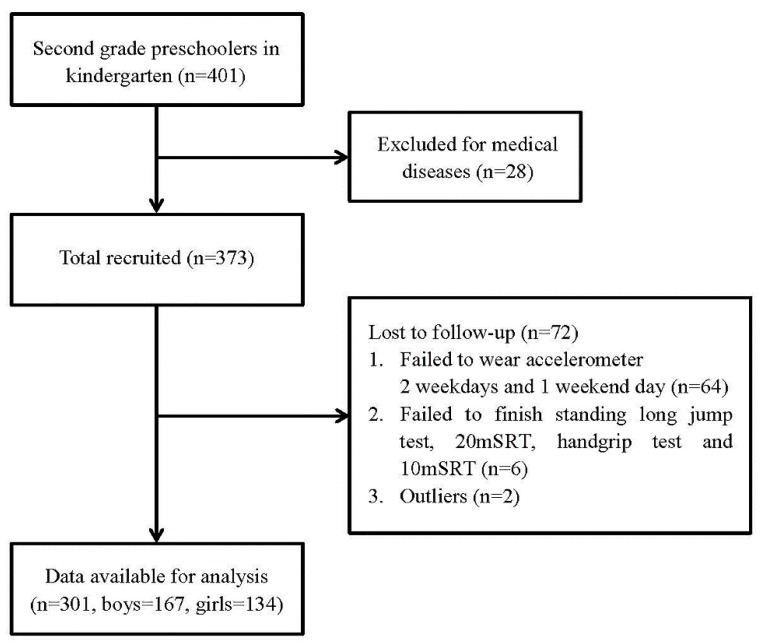
Flow of the subjects in the present study.

**Figure 2 jcm-09-00163-f002:**
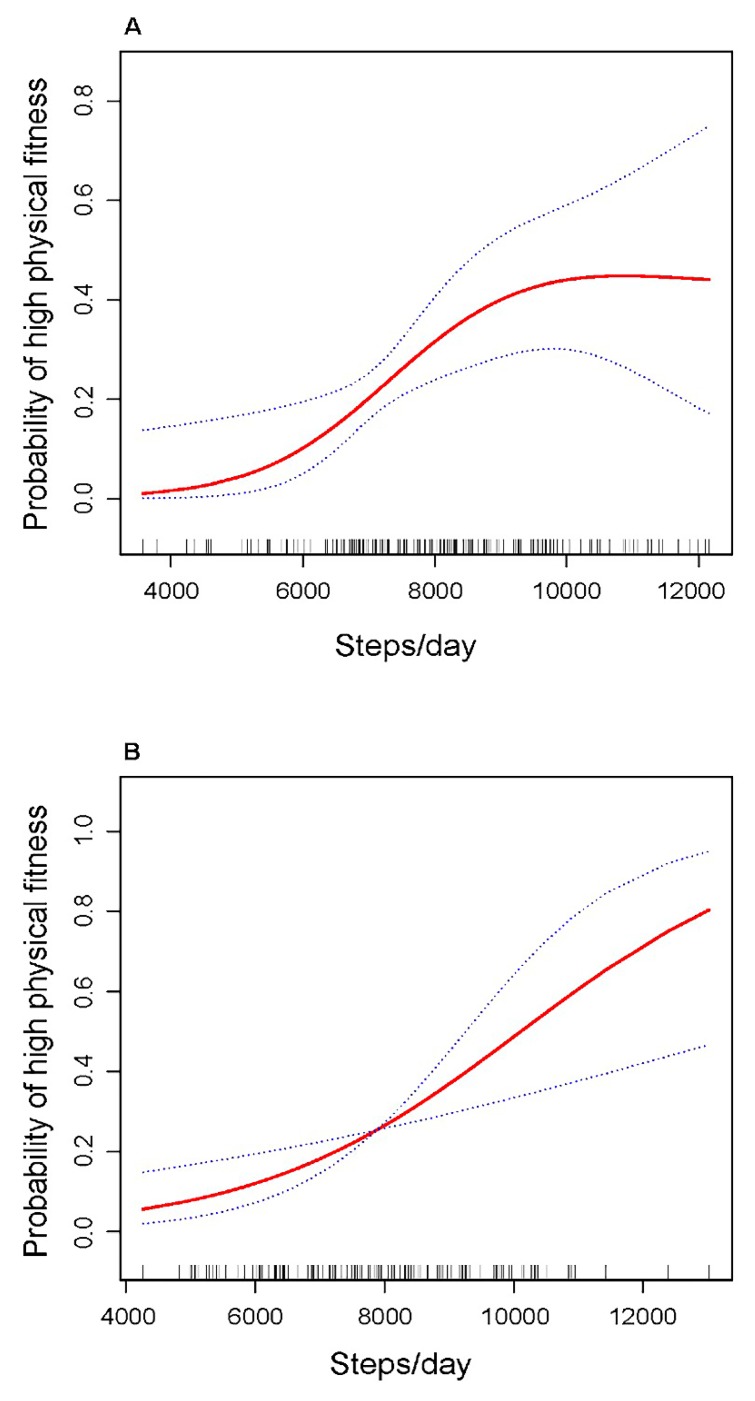
Associations between daily step count and high physical fitness level. ((**A**) for boys, (**B**) for girls; the y axis shows the probability of achieving the high physical fitness, the red solid line shows the fitted curves, the blue dot lines show the 95% confidence intervals. Both figures adjust for age (months), body mass index (kg/m^2^) and accelerometer wear time (min/day)).

**Table 1 jcm-09-00163-t001:** Characteristics of preschool children by daily step counts.

Characteristics	Steps per Day							
	Boys (*n* = 167)				Girls (*n* = 134)			
	T1 (3575–7168)	T2 (7169–8739)	T3 (8740–12162)	*p*-Value for Tertiles	T1 (4263–7047)	T2 (7048–8550)	T3 (8551–13026)	*p*-Value for Tertiles
Age (months)	58.45 ± 5.49	58.11 ± 5.57	57.73 ± 5.02	0.788	56.36 ± 5.61	57.00 ± 5.44	58.84 ± 5.14	0.072
Height (cm)	111.54 ± 4.82	111.59 ± 4.72	112.97 ± 5.26	0.125	109.06 ± 5.16	110.20 ± 4.85	112.75 ± 3.73 ^a,b^	**<0.001**
Weight (kg)	20.59 ± 3.49	20.02 ± 3.58	21.78 ± 3.67 ^a^	**0.008**	18.89 ± 2.80	18.95 ± 3.17	20.21 ± 2.18 ^a,b^	**0.002**
Body mass index (kg/m^2^)	16.49 ± 2.10	15.98 ± 1.89	16.98 ± 1.91 ^a^	**0.005**	15.86 ± 1.85	15.54 ± 1.81	15.90 ± 1.53	0.258
Standing long jump (cm)	84.69 ± 14.23	87.00 ± 18.13	85.45 ± 17.95	0.721	79.12 ± 15.28	84.23 ± 18.65	81.81 ± 15.41	0.251
Handgrip (kg)	6.83 ± 2.50	7.09 ± 2.14	7.41 ± 2.45	0.333	5.08 ± 2.14	6.24 ± 2.06^c^	6.87 ± 2.57 ^b^	**0.002**
2 × 10 m shuttle run (s)	7.23 ± 0.77	7.06 ± 0.74	7.11 ± 1.07	0.176	7.68 ± 0.72	7.09 ± 0.74^c^	6.92 ± 0.50 ^b^	**<0.001**
20 m shuttle run (laps)	12.48 ± 4.37	12.78 ± 4.62	12.54 ± 5.17	0.965	11.67 ± 3.21	13.91 ± 6.72	13.89 ± 4.80 ^b^	**0.040**
Accelerometer wear time (min/day)	703.63 ± 80.35	750.07 ± 66.64 ^c^	778.65 ± 63.61 ^a,b^	**<0.001**	713.67 ± 64.90	737.71 ± 78.29	764.97 ± 60.90 ^b^	**0.002**
High physical fitness								
No	50 (89.29%)	36 (65.45%)	39 (69.64%)		39 (86.67%)	33 (75.00%)	28 (62.22%)	
Yes	6 (10.71%)	19 (34.55%)	17 (30.36%)		6 (13.33%)	11 (25.00%)	17 (37.78%)	

Notes: Statically significant values are in bold. a: T3 versus T2, *p* < 0.05; b: T3 versus T1, *p* < 0.05; c: T2 versus T1, *p* < 0.05.

**Table 2 jcm-09-00163-t002:** Associations between daily step counts and physical fitness.

Steps/Day	*n*	Boys (*n* = 167) Odds Ratio (95% CI)		Steps/Day	*n*	Girls (*n* = 134) Odds Ratio (95% CI)	
		Model 1	Model 2			Model 1	Model 2
T1 (3575–7168)	56	1.0 (ref)	1.0 (ref)	Q1 (4263-6447)	45	1.0 (ref)	1.0 (ref)
T2 (7169–8739)	55	4.40 (1.60, 12.11)	5.53 (1.83, 16.72)	Q2 (6448-7865)	44	2.17 (0.72, 6.49)	2.44 (0.74, 8.09)
T3 (8740–12162)	56	3.63 (1.31, 10.08)	5.39 (1.65, 17.59)	Q3 (7866-9170)	45	3.95 (1.38, 11.27)	4.42 (1.30, 14.99)
*p* for trend		0.02	0.01	*p* for trend		< 0.01	< 0.02
Increase 1000 steps/day		1.32 (1.07, 1.62)	1.43 (1.10, 1.85)	Increase 1000 steps/day		1.50 (1.17, 1.93)	1.62 (1.20, 2.19)

Notes: Model 1 = no adjustment; Model 2: adjusted for age (months), body mass index (kg/m^2^), accelerometer wear time (min/day).

**Table 3 jcm-09-00163-t003:** Associations between daily steps counts and high physical fitness in boys (n = 167).

Steps (steps/day)	Odds Ratio (95%CI) for every 1000 Steps/Day Increment	
	Model 1	Model 2
Steps < 8000	2.31 (1.22, 4.37)	2.85 (1.29, 6.32)
Steps ≥ 8000	0.97 (0.68, 1.39)	1.04 (0.70, 1.55)
Likelihood Ratio	**0.033**	**0.035**

Notes: Model 1: no adjustment; Model 2: adjusted for age (months), body mass index (kg/m^2^), accelerometer wear time (min/day); Statically significant values are in bold.

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
