# Peer review of "Associations between Daily Step Counts and Physical Fitness in Preschool Children"

_jcm, 2020, doi:10.3390/jcm9010163_

Round 1

Reviewer 1 Report

The study is very interesting in the subject chosen and I understand that it can be a very interesting study for the scientific community.
The study comes with parts of it indicated in yellow and it seems that the work has been previously reviewed.

The introduction I think is quite interesting and the references that support it are appropriate. The material and methods are also successful and the pesentacion of the results is also perfect.
Properly presents the discussion, conclusions and limitations.
The article is fine for publication in the current state

Author Response

Responds to the reviewer 1 comments:

The study is very interesting in the subject chosen and I understand that it can be a very interesting study for the scientific community.

The study comes with parts of it indicated in yellow and it seems that the work has been previously reviewed.

The introduction I think is quite interesting and the references that support it are appropriate. The material and methods are also successful and the presentation of the results is also perfect.

Properly presents the discussion, conclusions and limitations.

The article is fine for publication in the current state.

Response: Thanks for your comments!

Reviewer 2 Report

Information for Authors:

General considerations:

Childhood sedentary lifestyle is associated with adverse health consequences, including atherosclerosis, hypertension, type 2 diabetes, fatty liver disease, and metabolic syndrome. Co-morbid health problems are common in obese children and include psychosocial disorders (e.g. depression, anxiety), skeletal disorders (e.g. musculoskeletal discomfort). Interest is growing in the connection between the decreased amount of daily physical activity, body fatness and children’s brain health, cognitive function and related attainments such as educational achievement and future socioeconomic success. The authors chose a very current topic. This problem was solved by a well-organized research strategy. The tasks were done with great humility to the problem.

Major revisions:

Please provide a more detailed description in the "Methods" chapter regarding the suitability of the motor tests in early childhood. Explain how informative the measure of (standing broad jump, handgrip)  force is for fitness.
Describe the criteria for performing the tests in detail! What type of instructions were communicated to the children?

Minor revisions:

In the chapter „Results” contains Table 1. Characteristics of preschool children by daily step counts. Make it clear where there are significant differences between fitness groups (T1, T2, T3). Describe what you mean (how define) a moderate-to-vigorous physical activity (MVPA).

Author Response

Responds to the reviewer 2 comments:

General considerations:

Childhood sedentary lifestyle is associated with adverse health consequences, including atherosclerosis, hypertension, type 2 diabetes, fatty liver disease, and metabolic syndrome. Co-morbid health problems are common in obese children and include psychosocial disorders (e.g. depression, anxiety), skeletal disorders (e.g. musculoskeletal discomfort). Interest is growing in the connection between the decreased amount of daily physical activity, body fatness and children’s brain health, cognitive function and related attainments such as educational achievement and future socioeconomic success. The authors chose a very current topic. This problem was solved by a well-organized research strategy. The tasks were done with great humility to the problem.

MAJOR REVISIONS:

Question 1:

Please provide a more detailed description in the "Methods" chapter regarding the suitability of the motor tests in early childhood.

Response: Thanks for your comment. The main components of motor fitness are speed, agility and balance, which was considered as one of major component of physical fitness in early childhood [1]. Motor fitness testing is also a very important testing item for Chinese National Preschool Children Physical Fitness Standard battery since 2003[2]. We revised in our manuscript accordingly, please see Page 4, Line 126-129.

Question 2:

Explain how informative the measure of (standing broad jump, handgrip) force is for fitness.

Response: Thanks for your comment. Musculoskeletal fitness (force) represents the capacity to carry out work against a resistance [3], it is significantly associated with adiposity and also a good predictor of CVD risk factors, premature mortality, and health outcomes in children, adolescents and adults [3-6]. Given its important to the overall fitness, musculoskeletal fitness has been included in several fitness-test batteries in children and adolescents [5, 7, 8]. In addition, previous study found that musculoskeletal fitness is a meaningful indicator of gross motor skills [9-11]. Motor skills which involving muscular strength are critical life skills [9-11], and were also found have links to cognitive development in preschool children [12]. Moreover, preschool children with better-developed motor skills may easier to be active and engage in more physical activity, which is critical to preschool children’s health development [13]. Therefore, it is necessary to measure musculoskeletal fitness when assess the physical fitness in preschool children [1].

Question 3:

Describe the criteria for performing the tests in detail! What type of instructions were communicated to the children?

Response: Thanks for your comment. We revised it in manuscript, please see Page 3, Line 107-112, Line 118-123, Line 126-129.

 MINOR REVISIONS:

Question 5:

In the chapter “Results” contains Table 1. Characteristics of preschool children by daily step counts. Make it clear where there are significant differences between fitness groups (T1, T2, T3).

Response: Thanks for your comment. We revised in manuscript, please see Page 4, Line 137; Page 6, Line 183-184.

Question 6:

Describe what you mean (how define) a moderate-to-vigorous physical activity (MVPA).

Response: Thanks for your inquiry. Generally speaking, moderate-to-vigorous physical activity (MVPA) refers to the activities with energy expenditure more than 3 METs [14, 15].

Reference:

Ortega, F.B., et al., Systematic Review and Proposal of a Field-Based Physical Fitness-Test Battery in Preschool Children: The PREFIT Battery. Sports Medicine, 2015. 45(4): p. 533-555. The National Physical Fitness Measurement Standards Manual (Preschool Children Version). 2003, People's physical education press, Beijing, China: The General Administration of Sport of China. Ortega, F.B., et al., Physical fitness in childhood and adolescence: a powerful marker of health. Int J Obes (Lond), 2008. 32(1): p. 1-11. Ruiz, J.R., et al., Predictive validity of health-related fitness in youth: a systematic review. Br J Sports Med, 2009. 43(12): p. 909-23. Ortega, F.B., et al., Muscular strength in male adolescents and premature death: cohort study of one million participants. BMJ, 2012. 345: p. e7279. Kemper, H.C., et al., A fifteen-year longitudinal study in young adults on the relation of physical activity and fitness with the development of the bone mass: The Amsterdam Growth And Health Longitudinal Study. Bone, 2000. 27(6): p. 847-53. Artero, E.G., et al., Reliability of Field-Based Fitness Tests in Youth. International Journal of Sports Medicine, 2010. 32(03): p. 159-169. Castro-Pinero, J., et al., Criterion-related validity of field-based fitness tests in youth: a systematic review. British Journal of Sports Medicine, 2009. 44(13): p. 934-943. Kim, C.I., D.W. Han, and I.H. Park, Reliability and validity of the test of gross motor development-II in Korean preschool children: applying AHP. Res Dev Disabil, 2014. 35(4): p. 800-7. Ulrich, D.A. and C.B. Sanford, Test of gross motor development. 1985: Pro-ed Austin, TX. Reeves, L., et al., Relationship of fitness and gross motor skills for five- to six-yr.-old children. Percept Mot Skills, 1999. 89(3 Pt 1): p. 739-47. Niederer, I., et al., Relationship of aerobic fitness and motor skills with memory and attention in preschoolers (Ballabeina): a cross-sectional and longitudinal study. BMC Pediatr, 2011. 11: p. 34. Williams, H.G., et al., Motor skill performance and physical activity in preschool children. Obesity (Silver Spring), 2008. 16(6): p. 1421-6. Organization, W.H., Global recommendations on physical activity for health. 2010. Health, U.D.o. and H. Services, 2018 Physical activity guidelines advisory committee scientific report. 2018.

This manuscript is a resubmission of an earlier submission. The following is a list of the peer review reports and author responses from that submission.

Round 1

Reviewer 1 Report

Introduction

The manuscript presents a correct structure presenting the most relevant points of the information, however it does not indicate the current status of the issue. Data such as similar studies are not indicated in this section.

Methods.

The Abstract indicates an amount that does not match the participants indicated in the method. What is the exact sample? 301 or 401?  Please Check this on the paper. You must to indicate in the abstact. It also does not indicate at what time the measurement was taken, nor was it done individually or in groups. What were children doing who were not being evaluated?

Furthermore the ethics committee number is not indicated. At the time they performed the strength test, was the grip length taken into account?

At what time were the tests performed? In the morning? after recess?

Correcting these aspects can enrich the manuscript

Reviewer 2 Report

MINOR REVIEW
Line 15. Where is boys mean age?

Line 27. Some keywords are repeated in the title

Line 97. SRT test was 2x10m, PREFIT recommends 4x10m, Why this change?

MAYOR REVIEW

Line 86. Why 2 + 1 days? the reference cited does not support this design, literal in conclusions page 489 (Penpraze, et al 2006): "The results of the present study suggest that a monitoring period of 7 days and 10 hr per day produced the highest level of reliability in measurements of physical activity in young children. However, surprisingly short monitoring periods may provide an appropriate level of reliability in this age group. These shorter monitoring durations need not include a weekend day. The evidence presented here should allow researchers to design practical research protocols at a specified level of reliability.
The authors assume 2+1 days like a proper protocol, but they did not calculate reliability with this protocol. In addition in this paper we can read: ”In the present study, reliability of measurement of physical activity became higher as the number of days of monitoring increased up to peak reliability of 80% (95% CI 71%, 86%) for 7 days. Whether the monitoring period included a weekend day did not significantly alter the reliability.
So the chosen protocol doesn't have reliability guaranteed

Line 149. Table 1.
These data are enlightening, mainly in boys:
1. There are no significant differences between the tertiles, in any physical test.
2. T2 is the best in three of the four physical tests performed
3. T3 has worn significantly more time the accelerometer, however, it was the best in only one test (handgrip performance has relation with step count?). More time with accelerometer more step counts?
Surprisingly, T3 has the best in the High physical fitness.

Line 151.
According to results showed in Table 1, T2 has the highest physical fitness, however T2 walked fewer steps than T3. But this results aren’t credible because the Accelerometer wear time is a bias and because the protocol aren’t reliable.

MAIN CONCERN
Lines 24-25. Establishing a cause and effect relationship between step counting and physical fitness is a big mistake. It is possible to find an association or a relationship, but never a cause-effect relationship with this research design.
The introduction is poor, the second question, specifically, is not correctly justified. The authors confuse MVPA, with physical fitness, are different concepts, and this is a bias that kept along the article.
It appears complicate justified scientifically that step count improve handgrip strength, for example.
In this sense:
Line 184-185. The second objective is inconsistent
Line 186. The authors state that there are a positive dose-response relationship between step counts and high physical fitness level. This paper doesn’t demonstrate this conclusion. There are not demonstrate a cause-effect relationship between step counts and physical fitness, maybe an association or a correlation, but never this research study dose-response.

Line 265-267. There are many methodologies and conceptual inconsistencies (explained before) to reach this conclusion